# HardCore Generation: Generating Hard UNSAT Problems for Data Augmentation

**Joseph Cotnareanu**[1,2,3,*]   **Zhanguang Zhang** [4]   **Hui-Ling Zhen**[4]
**Yingxue Zhang**[4]   **Mark Coates**[1,2,3,*]
McGill University[1], International Laboratory on Learning Systems (ILLS)[2],
Mila Québec Institute[3], Huawei Noah's Ark Lab[4]
`joseph.cotnareanu@mail.mcgill.ca,mark.coates@mcgill.ca,`
`{zhanguang.zhang,zhenhuiling2,yingxue.zhang}@huawei.com`

## Abstract

Efficiently determining the satisfiability of a boolean equation — known as the SAT problem for brevity — is crucial in various industrial problems. Recently, the advent of deep learning methods has introduced significant potential for enhancing SAT solving. However, a major barrier to the advancement of this field has been the scarcity of large, realistic datasets. The majority of current public datasets are either randomly generated or extremely limited, containing only a few examples from unrelated problem families. These datasets are inadequate for meaningful training of deep learning methods. In light of this, researchers have started exploring generative techniques to create data that more accurately reflect SAT problems encountered in practical situations. These methods have so far suffered from either the inability to produce challenging SAT problems or time-scalability obstacles. In this paper we address both by identifying and manipulating the key contributors to a problem's "hardness", known as cores. Although some previous work has addressed cores, the time costs are unacceptably high due to the expense of traditional heuristic core detection techniques. We introduce a fast core detection procedure that uses a graph neural network. Our empirical results demonstrate that we can efficiently generate problems that remain hard to solve and retain key attributes of the original example problems. We show via experiment that the generated synthetic SAT problems can be used in a data augmentation setting to provide improved prediction of solver runtimes[2].

## 1   Introduction

The boolean satisfiability problem (the SAT problem) emerges in multiple industrial settings such as circuit design (Goldberg et al., 2001), cryptoanalysis (Ramamoorthy and Jayagowri, 2023), and scheduling (Habiby et al., 2021). While machine learning is not well suited for solving SAT problems — solvers are typically required to have perfect accuracy and return correct proofs — it does have applications in predicting wall-clock solving time for a given solver, which is important for algorithm selection (Kadioglu et al., 2010; KhudaBukhsh et al., 2009) and benchmarking (Fuchs et al., 2023). SAT has also been gaining attention in Large-Language-Model reasoning, as it is a natural tool for interacting with the propositional-logical structure of many reasoning problems (Ye et al., 2023).

A major challenge for SAT-related learning is the scarcity of high quality, reasonably homogeneous, real-structured data. The most commonly-used datasets have been compiled via a series of annual

---

*We acknowledge the support of the Natural Sciences and Engineering Research Council of Canada (NSERC). Nous remercions le Conseil de recherches en sciences naturelles et en génie du Canada (CRSNG) de son soutien.

[2]We make our code publically available at `https://github.com/joseph-cotnareanu/HardCore`

International SAT Competitions. The industrial origins of the compiled instances differ substantially, so the dataset is highly heterogeneous. The data is a good test for heuristic SAT solvers but for data-driven learning methods, this heterogeneous, sparse data is unsuitable. More complex models are thus forced to use randomly generated data (Selsam and Bjørner, 2019). This is problematic because the hardness-inducing dynamics in industrial data are very different from those in randomly generated problems. Training or testing on most existing randomly generated data provides little insight into how a model will perform on real industrial problems (Balyo et al., 2022).

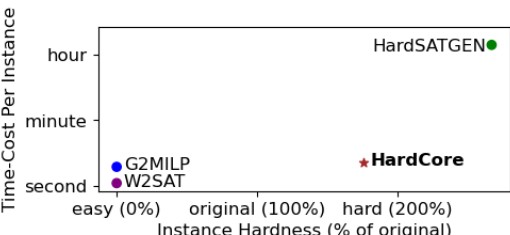

Figure 1: Our method (HardCore) achieves the best trade-off of inference cost and SAT-problem hardness.

Recently, deep-learning methods have been introduced to generate more realistic SAT instances. Early models (Wu and Ramanujan, 2019; You et al., 2019; Garzón et al., 2022) can generate instances that are structurally similar to original instances, but the problems are considerably easier to solve, a phenomenon called hardness collapse. Preserving hardness is essential, as generating only very easy problems renders the resultant dataset ineffective for distinguishing the best-performing solver from the worst. Additionally, such datasets fail to help the model learn to predict real runtimes. A recent study has succeeded in preserving hardness (Li et al., 2023). Unfortunately, the resultant method is prohibitively computationally expensive for synthetic data generation and augmentation for deep-learning. It can take over a week to generate a limited number of new problem instances. We summarize the cost/hardness trade-offs in Figure 1.

In this work, we take advantage of the connection between a problem's *core* and its hardness. The core is comprised of the identifiable minimal subsets of a boolean SAT problem that are unsatisfiable (UNSAT). Our strategy is to preserve the core of an original instance while iteratively adding random clauses to construct similar, but sufficiently diverse, problem instances that can enhance learning. To do this, we need to detect the core after each iteration. Unfortunately, traditional core detection algorithms are slow and can take hundreds of seconds, as they often require to solve the SAT problem (Wetzler et al., 2014). Clearly, such an algorithm is impractical for building a fast generator, as core detection needs to be performed hundreds of times for every instance we generate.

To address this, we rephrase core detection as a binary node classification algorithm (core/not-core). We train a graph neural network to perform the task. Importantly, we can circumvent the data starvation issue, because our random data generation procedure generates hundreds of example instances that can be used for training the core detection algorithm. We can also take advantage of the fact that while it is important to identify the vast majority of clauses that belong to the core, we can tolerate a relatively high number of false-alarms by post-processing with a fast pruning algorithm.

We make the following novel research contributions:

- We propose a novel method for SAT generation that is the first that can both (i) *preserve hardness* and (ii) *generate instances in a reasonable time frame*. We can thus generate thousands of hard instances to augment a dataset in minutes or hours.

- We demonstrate experimentally that our proposed procedure preserves the key aspects of the original instances that impact solver runtimes. This hardness preservation is crucial when augmented dataset is used to learn to predict solver times, a vital task for solver benchmarking and selection.

- We illustrate the value of our augmentation process for solver runtime prediction. On an example dataset, our augmentation process reduces mean absolute error by 20-50 percent. In contrast, all other generation algorithms achieve no statistically significant improvement.

## 2   Background: Boolean Satisfiability

**Definitions and Notation**   The Boolean Satisfiability Problem (SAT) is the problem of determining whether there exists an assignment of variable values that satisfies the given Boolean formula, rendering it true. Typically, a SAT instance is represented in Conjunctive Normal Form (CNF), which is written as a conjunction (logical AND) of disjunctions (logical OR), for example $f =$

$(\neg A \lor B \lor C) \land (A \lor \neg C) \land (\neg B \lor C)$. The signed version of each variable that appears in the formula is known as a literal. For example, $A$ and $\neg A$ are both literals of the variable $A$ (Biere et al., 2009, Chapter 2).

Another useful representation of a CNF is as a set of sets, where each set (referred to as a clause) represents a disjunction in the CNF and contains the literals included in that disjunction. Denote the $i$-th clause in the formula $f$ by $c_i$ and the $j$-th literal in clause $c_i$ as $l_j$. If there are $n_c$ clauses in $f$ and $n_{l_i}$ literals in clause $c_i$, we can express the formula as $c_i = \bigcup_{j=1}^{n_{l_i}} l_j$, $f = \bigcup_{i=1}^{n_c} c_i$.

**Core Definition**   Given an unsatisfiable (UNSAT) instance $U$, there is a subset of clauses called a Minimally Unsatisfiable Subset (MUS) or a Core. This subset is the smallest possible subset of clauses from $U$ that is UNSAT (Biere et al., 2009, Chapter 11).

**Graph Representation of CNFs**   There are several common CNF graph representations. In this work, we use the Literal-Clause Graph (LCG), an undirected and bipartite graph. Each node in the first set of nodes represents a clause and each node in the second represents a literal. We construct an edge for each occurrence of a literal in a clause; the set of undirected edges $e$ is defined as $e = \bigcup_{i=1}^{n_c} \bigcup_{j=0}^{n_{l_i}} (l_{j_{c_i}}, c_i)$.

# 3   Related Work

## 3.1   Deep-learned SAT generation

The problem of learned generation for SAT problems was first established in 2019 with SATGEN (Wu and Ramanujan, 2019), motivated by a lack of access to industrial SAT problems. SATGEN used a graph generative adversarial network (GAN) to generate graph representations of SAT problems.

G2SAT (You et al., 2019) represents problems as graphs. The graphs are progressively split into small trees, and a graph neural network (GNN) is trained to discern which trees should be merged to restore the original graph. While innovative, the method is slow due to its need to sample many tree pairs to form a SAT problem of sufficient size. The most recent improvement on the G2SAT framework, HardSATGEN (Li et al., 2023), includes some domain-inspired considerations in its design, such as communities and cores. HardSATGEN is the first deep-learned SAT generation method that can generate problems which are not trivial to solve for solvers: often the generated problems take nearly as long or even longer for a solver to solve than the corresponding seed problem. Unfortunately, however, the core awareness aspects of the design cause HardSATGEN to be extremely slow, making it challenging to use in any setting that needs many new instances.

W2SAT (Wen and Yu, 2023) follows an approach more similar to the original SATGEN. It employs a low-cost general graph generation model, and obtains new SAT problems via graph decoding. W2SAT is extremely efficient, but like G2SAT, it is incapable of generating hard problems. G2MILP (Geng et al., 2023). is designed to generate Mixed Integer Linear Programs (MILPs), which are the general case of SAT. A naive modification allows us to use G2MILP to generate SAT problems. The method is nearly as efficient as W2SAT, but also struggles to generate hard instances.

## 3.2   Core Prediction

Core Detection can be a helpful tool for understanding UNSAT problems. Cores are often seen as a strong indicator of the hardness of an UNSAT problem (Ansótegui et al., 2008). There are multiple classical, verifiable methods for Core Detection, with the current standard being Drat-Trim (Wetzler et al., 2014). Drat-trim requires that the problem be solved once by a SAT solver, which is very slow. In response to this, Neurocore (Selsam and Bjørner, 2019) was designed to predict the core of a SAT problem. Neurocore converts the input problem to a graph and uses a GNN to predict cores. Strangely, however, Neurocore does this on variables rather than clauses. Cores are defined to be subsets of clauses, rather than variables, and so this choice seems unnatural. Neurocore strives to be a machine-learning based variable-selection heuristic for SAT solvers, which motivates the focus on variables.

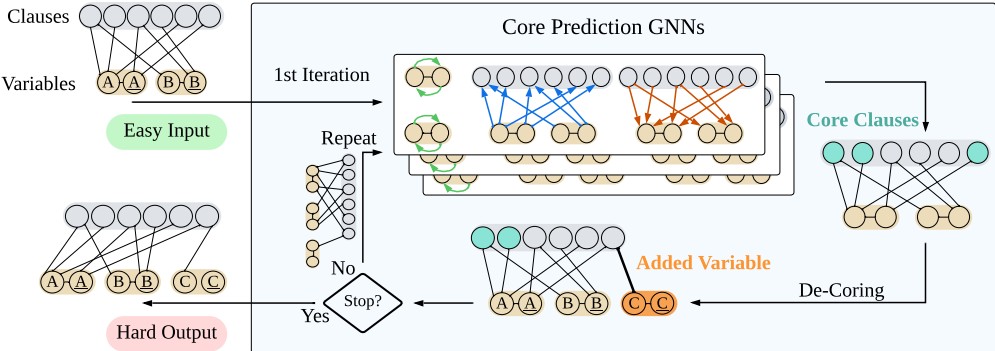

Figure 2: **Core Refinement**. The core refinement process comes in two steps: (1) Core Prediction, in which we use a GNN-based architecture to identify the core of the generated instance; and (2) De-Coring, in which we add a non-conflicted literal to a clause in the core, rendering the core satisfiable and giving rise to a new, harder minimal unsatisfiable subset (core). As steps (1) and (2) are repeated, the easiest core of the problem is gradually refined, raising the hardness of the generated instances.

## 4 Problem Statement

Given a training set of UNSAT CNFs $S = \{f_1, f_2, ..., f_{m_S}\}$, and a corresponding set of label vectors $R = \{\mathbf{r}_1, \mathbf{r}_2, ..., \mathbf{r}_{m_S}\}$, we wish to train a generative model $G$ that can construct new examples. The label vector $\mathbf{r} \in \mathbb{R}^d$ represents the hardness of the SAT problem and we model it as a deterministic mapping, i.e., $\mathbf{r}_1 = g(f_1)$. In our experiments, the vector is derived by recording the SAT solving time for a pre-specified set of SAT solvers.

We assume that the $m_S$ CNFs in the training set are i.i.d. examples from an underlying distribution $\mathcal{D}$. We denote the generative model distribution by $\mathcal{D}_G(S)$, highlighting that it is dependent on the random training set $S$. We can obtain a new dataset of $m_G$ i.i.d. samples $S_G$ using the generative model. The total number of samples in the augmented set $\tilde{S}$ is then $m_S + m_G$.

Our primary goal is to derive a generative procedure that produces sufficiently representative but also diverse samples such that the error obtained by training a model on the augmented dataset $\tilde{S}$ is less than that obtained by training on the original dataset $S$. As an example task, we consider the prediction of runtime for a candidate solver. In this case, the appropriate loss function is the absolute error between the predicted time and the true time.

Beyond this, we are also interested in the distance between the distributions $\mathcal{D}$ and $\mathcal{D}_G$. We examine this through the lens of hardness label vectors. The application of $g$ to the CNF descriptors generated according to $\mathcal{D}$ or $\mathcal{D}_G$ induces distributions in $\mathbb{R}^d$. To evaluate the similarity of the original and generated instances, we calculate the empirical maximum mean discrepancy (MMD) distance between these induced distributions.

## 5 Methodology

Our generation strategy can be broken into three steps: (1) extraction of the core from a seed instance; (2) addition of random new clauses, generated with low cost; and (3) iterative core refinement. Figure 2 provides an overview of the key core refinement procedure. It consists of a two-step cycle of (a) high-speed core extraction using our novel GNN-based method; and (b) unconflicted literal addition to break any undesirably easy core.

### 5.1 Generating Hard Instances

**Trivial Cores** Cores are the primary underlying hardness providers in UNSAT instances, because a solver must only determine that a subset of a CNF is UNSAT for the whole CNF to be UNSAT, and

a core is the smallest subset of clauses of a CNF that is UNSAT. Small cores with few clauses are generally easier to solve due to less variable assignment combinations. An example of a trivial core is $(A \vee B) \wedge (\neg A \vee B) \wedge (A \vee \neg B) \wedge (\neg A \vee \neg B)$.

Whenever we add a new random clause to an UNSAT instance, there is the danger of creating a trivial core. For example, consider an UNSAT instance which includes three of the clauses from the example above: $(A \vee B) \wedge (\neg A \vee B) \wedge (A \vee \neg B)$. If during generation we unknowingly add the clause $(\neg A \vee \neg B)$, the UNSAT instance's large (hard) core will be replaced by a trivial one, leading to hardness collapse. Maintaining awareness of cores and potential cores in a CNF as we perform modifications is challenging. We take a different approach, which we refer to as *Core Refinement*.

**Core Refinement**  The Core Refinement process is made up of two steps that are repeated $n$ times, where $n$ is the number of generated clauses. The procedure is depicted in Figure 2. The first step of the process is to identify the core of the generated instance. The addition of random clauses during generation is very likely to create a core that is trivially easy to solve and it may not be the same as the core of the original instance. Once we have detected this easy core, we make it satisfiable by adding a new literal to a clause in the core. The addition of a single, flexible literal eliminates the constraints of the core and makes it possible to satisfy.

Returning to the previous example, the UNSAT CNF $(A \vee B) \wedge (\neg A \vee B) \wedge (A \vee \neg B) \wedge (\neg A \vee \neg B)$ can be made satisfiable by modifying any of the clauses in this fashion: $(A \vee B \vee C) \wedge (\neg A \vee B) \wedge (A \vee \neg B) \wedge (\neg A \vee \neg B)$. The introduction of literal $C$ in the first clause means that $(A = 0, B = 0, C = 1)$ is now a satisfying solution.

As these two steps are repeated, the core of the instance gradually becomes larger and is likely to be more difficult. The process ends after a fixed number of iterations. In our experiments, we choose this to be the number of generated clauses. Since the hardness of the core is the hardness of the instance (Ansótegui et al., 2008), the refinement process can be seen as progressively raising the hardness of the problem.

Figure 3: **Core Prediction GNN Architecture**. We construct our GNN using three parallel message passing neural networks (MPNN) whose calculated node embeddings are aggregated at each layer to form the layer's node embeddings. Readout is done by taking the sigmoid of a fully-connected layer on clause node embeddings and thresholding. Training is supervised by taking a binary classification loss between the true core labels and the clause nodes' core prediction probabilities.

**Underlying Hard Core Guarantee**  The Core Refinement process is designed to repeatedly eliminate easy cores, so after each iteration, the core becomes harder. Finally, after many iterations, we hope that the remaining core is as hard as the original instance. This process can only be guaranteed to lead to a hard core if an underlying hard core exists in the instance at the start of the refinement process. Refinement then whittles away easy cores until only the hard one remains.

There is a possibility of creating a hard core through the random generation of clauses, but we cannot rely on this. We must introduce an element to our design to ensure there is a hard core. To achieve this we identify cores from the original instances and include them in the generated instances.

## 5.2  Core Prediction

We have two critical objectives for our method: low cost and hard outputs. While the Core Refinement process serves us well in generating hard instances, a naive implementation using existing core detection algorithms is unacceptably expensive in terms of computation requirements. Current core

detection algorithms first solve the SAT problem, making Core Detection NP-Complete (Wetzler et al., 2014).

We adopt the strategy of approximating the Core Detection algorithm. Since an instance can be naturally represented using a bipartite graph, and the goal of core detection is binary classification of each clause, we expect that a graph neural network is a promising approach.

**Graph Construction**    We represent each instance as a graph as outlined in Section 2. We make two changes: (a) we add message-passing edges to connect matching positive and negative literals (e.g., $\neg A$ and $A$); (b) we replace each undirected edge with two directed edges. These changes are designed to facilitate the diffusion of information in the GNN. We denote the set of literal-literal message passing edges by $\mathcal{E}_{ll} = \bigcup_{i=1}^{n_v}(l_{i+}, l_{i-})$, where $n_v$ is the number of variables in the instance. We denote the set of literal-to-clause directed edges by $\mathcal{E}_{lc} = \bigcup_{i=1}^{n_c} \bigcup_{j=0}^{n_{l_{c_i}}}(l_{j_{c_i}}, c_i)$. We denote the set of clause-to-literal directed edges by $\mathcal{E}_{cl} = \bigcup_{i=1}^{n_c} \bigcup_{j=0}^{n_{l_{c_i}}}(c_i, l_{j_{c_i}})$.

**GNN Architecture**    Given the heterogeneous nature of our graph, arising from different node and edge types, we use three Graph Message Passing models (one for each edge type), as described in Figure 3. We couple these models by averaging their embeddings after each layer. We define a single layer where $\sigma$ is a non-linear activation function. Finally, we obtain a core membership probability for each clause node by passing the embeddings through a fully connected linear readout layer followed by a sigmoid function to the clause node embeddings. We threshold the values to obtain positive and negative classifications of core membership:

$$h^{l+1} = \sigma(\frac{1}{3}(GNN(\mathcal{V}, \mathcal{E}_{cl}, h^l) + GNN(\mathcal{V}, \mathcal{E}_{lc}, h^l) + GNN(\mathcal{V}, \mathcal{E}_{ll}, h^l))), \qquad (1)$$

$$out = \mathbb{1}_{>0.5}(\sigma(xh_c^L + b)). \qquad (2)$$

**Training**    Our augmentation process is motivated by a scarcity of data. We must therefore address this when training the core detection GNN. We achieve augmentation of the available data by executing the generation pipeline described above for a small number of instances, using a slow, traditional but proof-providing tool for Core Detection in the Core Refinement process. By saving the instance-core pair after each iteration of the core refinement process, we can construct sufficient supervision data for training the Core Prediction GNN model. Although the instance-core pairs we construct this way are correlated, there is sufficient variability for the GNN model to generalize well to other instances. We train the model using the standard binary cross-entropy loss function. For experimental results showing the performance of our Core Prediction model, see row titled "LEC" in Table 4 in the Appendix B.

## 6    Experiments and Results

### 6.1    Experimental Setting

**Proprietary Circuit Data (LEC Internal)**    This LEC Internal data is a set of UNSAT instances which are created and solved during the Logic Equivalence Checking (LEC) step of circuit design. LEC needs to be performed after certain circuit optimization steps to ensure that the optimization process has not corrupted the logic of the circuit. If the logic is uncorrupted, the created SAT problem will be UNSAT. Since it is extremely rare that these optimizations in fact corrupt the circuit, more than 99% of LEC instances are UNSAT. For each generative method, we will generate 5 problems for each problem in the data.

**Synthetic Data (K-SAT Random)**    Acknowledging the importance of reproducibility, we also provide results on synthetic data. This data is generated by randomly sampling a CNF with $m$ clauses of $k$ literals over $n$ variables. Clauses are sampled without replacement. We have previously argued that random data differs from real data in important ways that make it unsuitable for machine learning applied to real problems. Holding to this view, we use this data primarily to provide a surrogate to the internal data for experimental reproduction purposes, rather than to present results on a second dataset. For each generative method, we will generate 5 problems for each problem in the data. For details concerning both the LEC Internal and K-SAT data, see Table 3 in the Appendix.

**Note on Public Competition Data**   A large and commonly-used public dataset for SAT is the SAT Competition data (Heule et al., 2019). Unfortunately, as argued previously, this data is generally ill-positioned for machine learning as it is highly heterogeneous. Despite this, we recognize its value and importance as a widely accepted dataset. Thus, we include in Tables 5 and 6 of Appendix B final results on the Data Augmentation experiment done below, done on the "Tseitin" and "FDMUS" families found in the public competition data. Our method shows improvement on the un-augmented data consistent with our findings on the synthetic and proprietary data prosented in this section.

**Data splits for training the HardCore GNN and the runtime-prediction model**   There are three separate groupings of the dataset: (i) Core Prediction training data, (ii) generation seeding data, and (iii) the remaining data. This split is chosen randomly. Core Prediction training data can be small (we used 15 problems), because we use each problem as a seed instance 5 times for generation followed by core-refinement with a traditional core detector. Saving problem-core pairs at each step, we obtain 15,000 training pairs for the core-predictor model. The seeding data are used to seed HardCore once the core predictor is trained in order to obtain generations to evaluate. These generations are then compared against the seed data for runtime similarity. Finally, these generations (and their seeds) are used to train a runtime-predictor model, which is evaluated on the remaining un-used data.

**SAT Solvers**   We select 7 solvers for hardness analysis: Kissat3 (Biere et al., 2020), Bulky (Fleury and Biere, 2022), UCB (Cherif et al., 2022), ESA (Cherif et al., 2022), MABGB (Cherif et al., 2022), moss (Cherif et al., 2022) and hywalk (Chowdhury, 2023). These solvers exhibit complementary performance characteristics: when some of these solvers perform well on certain instances, some perform very poorly. This results in a diverse runtime distributions in our analysis. We run our experiments on a Intel(R) Xeon(R) Platinum 8276 CPU @ 2.20GHz cpu and 3 Nvidia Tesla V100 GPUs.

We compare to the following baselines:

- **HardSATGEN (Li et al., 2023)**: A high-cost split-merge generator with community structure and core detection that is capable of generating hard instances.
- **W2SAT (Wen and Yu, 2023)**: A low-cost generative method that utilizes a less common SAT graph representation which was reported to generate very easy problems.
- **G2MILP (Geng et al., 2023)**: A low-cost VAE-based generative model designed for the general case of SAT: MILPs.

## 6.2   Research Questions

Our work is motivated by the goal of **fast** generation of **hard** and **realistic** UNSAT datasets for **data augmentation**. Given these goals, we now establish our strategy for evaluating our model, identifying the key research questions that our experiments explore.

### 6.2.1   Question 1: Is the method able to generate hard instances?

In order to quantify 'hardness', we choose the wall-clock solving time for each solver as a metric. We deem a set of generated instances 'hard' if the average solver runtime is at minimum 80% of the original dataset's average hardness. If average solver time for the set of generated instances is below 5%, we consider that *hardness collapse* has occurred.

In Table 1 we compare generated with original hardness. W2SAT and G2MILP both suffer hardness collapse, whereas HardSATGEN and HardCore generate hard instances.

### 6.2.2   Question 2: Is the method fast?

We measure generation speed by the time required to generate an instance (in seconds). We evaluate this by measuring the wall-clock time of each model during inference and dividing by the number of generated instances. Generally, a method should be able to generate hundreds of instances per hour so that we can augment a dataset in a reasonable time frame.

In Table 1, the division between fast and slow procedures is very clear: W2SAT, G2MILP, and HardCore all exhibit similar instance generation times, with W2SAT being the fastest. In contrast,

Table 1: Evaluation of generated datasets on LEC data. Hardness level (%): percentage of runtime of generated dataset relative to original dataset, closer to 100% is better. Speed (s): average time cost to generate one instance, lower is better. Maximum Mean Discrepancy (MMD): distance between distributions of generated and original datasets, lower is better.

|                       | W2SAT    | HardSATGEN | G2MILP   | HardCore  |
| --------------------- | -------- | ---------- | -------- | --------- |
| Hardness (%)          | $\sim 0$ | 267        | $\sim 0$ | **176**   |
| Time per instance (s) | **1.2**  | 6441       | 3.3      | 4.3       |
| Similarity (MMD)      | —        | 0.492      | —        | **0.004** |

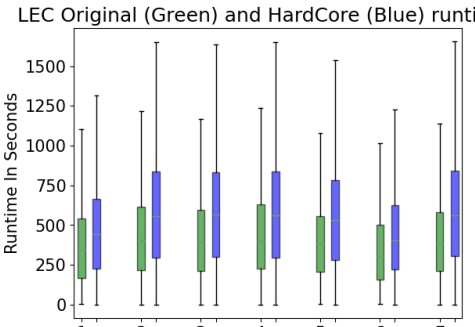 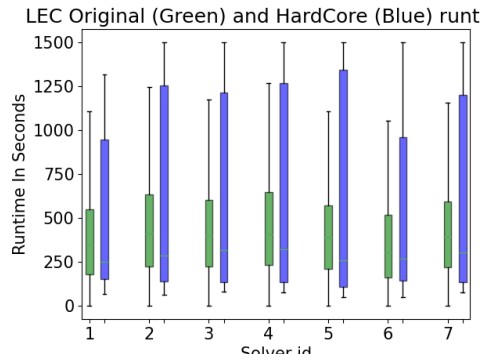

Figure 4: HardCore (Left) and HardSATGEN (Right). Boxplots of runtimes per solver for Original (Green) and Generated (Blue) instances on LEC data. HardCore appears to produce per-solver distributions which are much closer to the original than HardSATGEN, which tends to produce high-variance and on-average much harder problems than the original.

HardSATGEN takes close to 2 hours to generate a single instance. To generate 1000 LEC instances at this speed we would need 75 days.

### 6.2.3   Question 3: Is the method able to generate datasets that are similar to the original datasets in terms of hardness distribution?

Although past work such as Li et al. (2023); Wen and Yu (2023); You et al. (2019) has examined graph statistics such as modularity and clustering coefficients, we find little evidence that these are indicative of the hardness of generated instances. Instead, we focus on the similarity of the distributions of the hardness vectors because hardness is of primary importance when working with SAT problems.

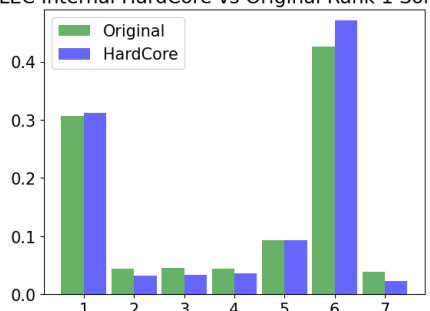 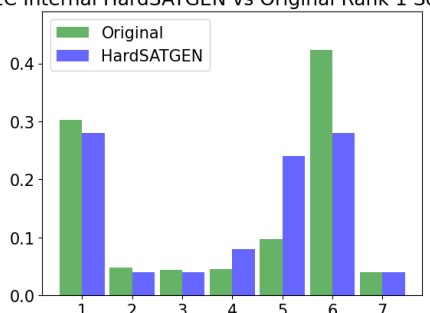

Figure 5: LEC Internal Rank 1 Solvers. We compare original and synthetic best-solver observations for HardCore (left) and HardSATGEN (right).

As G2MILP and W2SAT exhibit hardness collapse, we only compare HardSATGEN and HardCore for runtime distribution analysis. Note that due to HardSATGEN's high cost, we can only generate 50 LEC instances and 50 K-SAT instances within 3 days. In the following experiments, we compare "original" and "generated" data. Here, "original" refers to only those instances used as seeds during inference for each model; "generated" refers to the outputs. Hence, the "original" sets for HardSATGEN and HardCore are different because the number of seed instances is different (due to time constraints we are limited in how many HardSATGEN instances we can generate). We evaluate the similarity between original and generated data through the Maximum Mean Discrepancy (MMD) metric, the runtime distribution, and the best solver distribution.

As shown in Table 1, HardCore achieves runtime distributions far closer to the original distributions compared to HardSATGEN with respect to the MMD metric. We calculate these values by taking the MMD between the set of instances used as seeds during generation (a subset of the training set) and the corresponding set of generated instances. We note that while HardCore achieves low MMD, the solving time of individual instances is considerably different from that of their associated seeds. This implies that low MMD of HardCore is not achieved by replicating or barely modifying seed instances. Our later experiments investigating augmentation suggest that there is sufficient diversity being injected in the generated instances.

In Figure 4, we visually compare the per-solver runtime distribution of HardCore's generated datasets to the corresponding original datasets. HardCore produces per-solver distributions which are visibly much closer to the original distributions than HardSATGEN. In Figure 5, we see a striking similarity between the HardCore distribution of best-performing solvers and the original distribution, indicating that the HardCore synthetic instances are solved most efficiently by the same solvers as the original instances, in a distributional sense. Meanwhile, a greater discrepancy can be seen between original and HardSATGEN-generated data, particularly for solvers 5 and 6. A full histogram of LEC solver ranks is shown in Figure 6 of Appendix B.

#### 6.2.4 Question 4: Can we successfully augment training data with the method's generated data for machine learning?

We address the task of runtime prediction and compare the performance of two models: one trained on only original data and the other trained on a dataset augmented with generated instances. We train the SATzilla model to predict solver runtime of one specific solver on a given instance. We repeat this for each of the 7 solvers. We calculate the MAE of the predicted total runtime for each solver and average over the solvers. We compare HardCore, W2SAT and two versions of HardSATGEN: (i) HardSATGEN-Strict and (ii) HardSATGEN-$N$. For HardSATGEN-Strict, we only generate as many instances as possible in the time it takes HardCore to generate the desired number of instances. For HardSATGEN-$N$, we generated $N$ instances, where $N$ was selected as the number that could be generated in approximately 3 days of computation. We also compare to the un-augmented training sets and refer to it as Original.

In order to observe performance over varying sizes of training data, we conduct this experiment for several quantities of original training instances, which is denoted Data Size. Three augmentation instances are generated per original instance, and augmentation is only allowed by using the original instances in the training set. Validation sets are selected from the original data only, with an 80/20 split train/validation split. For LEC the test-set is made up of 10000 randomly selected problems which were not selected for training or validation. For K-SAT the test-set is made up of the problems which were not picked for train/validation from the 1351 original instances.

Table 2 shows that for both K-SAT random data and LEC Internal dataset, training on data augmented using HardCore leads to a 20-50 percent reduction in MAE. The gain of data augmentation increases with larger data size. In contrast, no other data generation method leads to a comparable improvement.

### 6.3 GNN generalization to other datasets: do we have to re-train for new unseen circuits?

In order to measure GNN generalization to new data without re-training, we create a new split of the LEC data. Each problem in the LEC data can be traced back to one of 29 circuits. By randomly splitting circuits into training circuits and test circuits (and then building training and evaluation sets with their respective problems), we can measure generalizability. Note that we would not expect the

Table 2: MAE of Runtime Prediction averaged across 7 solvers and 15 trials. Asterisks are placed at the best result which passes the Wilcoxon pairwise ranking test against the second-best for $p < 0.05$. For a boxplot visualization showing each trials result, see Appendix Figure 7

|  | K-SAT Random | | | | LEC Internal | | | | |
| --- | --- | --- | --- | --- | --- | --- | --- | --- | --- |
| Data Size | 10 | 20 | 30 | 40 | 100 | 200 | 300 | 400 | 500 |
| HardSATGEN-$N$ | 2416 | 2306 | 2172 | 2182 | 666 | 797 | 605 | 617 | 463 |
| HardSATGEN-Strict | 2179 | 2578 | 2488 | 2456 | 627 | 742 | 565 | 638 | 513 |
| W2SAT | 2606 | 2046 | 1807 | 1377 | 724 | 704 | 634 | 611 | 535 |
| Original | 2750 | 2743 | 2109 | 1449 | 707 | 795 | 557 | 606 | 526 |
| HardCore | **2156** | **1796*** | **1615** | **930*** | **514** | **481*** | **369*** | **282*** | **338*** |

model to generalize to problems derived from a completely different application domain (although fine-tuning a previously model in a domain adaptation strategy might be interesting to explore).

In row titled "Circuit-Split LEC" in Table 4 of Appendix B we report the GNN performance on this experiment. In the paper we discussed that Core recovery is the priority, because if we falsely classify true-positives then we may be unable to de-core the current core (since the necessary clause may be undetected, whereas if we mis-classify true-negatives then we will simply de-core a non-core clause. Given enough iterations of core-refinement, a true-positive clause will eventually be selected for de-coring (since the clause for de-coring is randomly selected from among the detected clauses). With this in mind, the threshold hyper-parameter which is used on the sigmoid outputs at model readout becomes a useful parameter in cases where classification performance is weakened: we can boost Core Recovery (recall) by lowering the threshold. Tuning this threshold is very low-cost: testing a thousand problems takes 500 seconds on a GPU. We find that by testing values $[0.1, 0.3, 0.5, 0.7, 0.9]$ — which takes 25 minutes — we can tune the threshold to provide similar recall to the in-distribution model.

# 7 Limitations

The primary limitation of our work is that it is restricted to UNSAT problems. While some SAT applications are almost entirely UNSAT (e.g., circuit design), many are not. With our proposed approach, this limitation is unavoidable because cores are only present in UNSAT problems. However, there is a concept for SAT problems analogous to the core, known as a *backbone*. Focusing on preserving the backbone is a potential avenue for an analogous method for satisfiable SAT problems.

Another limitation is that our work relies solely upon empirical results to demonstrate its efficacy, and these results are only presented on two datasets, one of which is synthetic. To partially address this concern, we conducted several trials and statistical significance testing to ensure the reliability of our empirical analysis.

Another limitation is that our method struggles to scale to extremely large SAT problems. As the size of the SAT problem increases, memory and computation costs scale in polynomial complexity, meaning that SAT problems which have millions of clauses are currently out of reach for this method. For a more in-depth discussion of scaling, please see the Scaling subsection A.3 in Appendix A.

# 8 Conclusion

We present a fast method for generating UNSAT problems that preserves hardness. Existing deep-learned SAT generation algorithms either (1) are incapable of generating problems that are even 5% as hard as the example input problems; or (2) can generate hard problems but take many hours for each instance. Our proposed method targets the core of a SAT problem and iteratively performs refinement using a GNN-based core detection procedure. Our experiments demonstrate that the method generates instances with a similar solver runtime distribution as the original instances. For a more challenging industrial dataset, we show that data augmentation using our proposed technique leads to a significant reduction in runtime prediction error.

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

# A  Appendix

## A.1  Data

Table 3: Data Statistics. Note that LEC is a much larger dataset than Tseitin in every regard: average variable and clause counts, average hardness on Kissat solver and dataset size.

|       | var. | clause | runtimes (s) | count |
|-------|------|--------|--------------|-------|
| LEC   | 1328 | 5167   | 388          | 78730 |
| K-SAT | 398  | 1751   | 2700         | 1351  |

## A.2  Hyper-parameters

In our design process, given the cost of running experiments — in particular measuring runtime of generated instances — we did not conduct exhaustive hyperparameter searches. Hyperparameters were set following design considerations and rationales, which will be discussed here.

- The random generation method we use is Popularity-Similarity. This has several hyper-parameters: average clause size, $\beta_c$, $\beta_v$ and $T$. Average clause size determines the average number of literals per generated clause, $\beta_c$ and $\beta_v$ are constants in the probability distribution for clause and variable selection, respectively, and $T$ is a constant in the exponent of the probability of an edge existing between clause and variable. Conducting an exhaustive search over these hyperparameters is expensive because the evaluation of each configuration is via runtime-measurement, which requires the solving of a large number of SAT problems by multiple solvers. We communicated with the authors of the paper which presented HardSATGEN, and were able to obtain their hyperparameter configuration for Popularity-Similarity (PS), which was included among their reported baselines. For continuity with previous work and in the interest of reducing the computational budget, we used the provided configuration.

- The GCN backbone within our core prediction module has two hyperparameters, namely the number of hidden dimensions and the number of layers. Three potential values were chosen for initial exploration of layer size: [3, 4, 15]. In many applications, GCN networks are configured to have only 3 or 4 layers. This is because GNN networks in general are prone to over-smoothing as the number of layers increases. 15 layers was added to validate this behavior within our context. For hidden dimension size we chose two potential values: [32, 64]. Our findings were that as the model size increased via additional layers and hidden feature size, there was minimal improvement in performance. Thus, we selected the smallest defined configuration of 3 layers and hidden dimension of 32.

- Finally, there is the Core-Refinement hyperparameter that specifies the number of iterations. This value can be set in terms of the number of generated clauses, since one clause is modified at each iteration. The safest setting is to set the number of iterations to be equal to the number of generated clauses, such that, if necessary, the method is allowed to modify every generated clause. In practice, this was the setting we used.

## A.3  Scaling

Memory limitations are the primary challenge for our proposed method, due to the need to store the graph. For our experimental hardware (32GB GPU) and our implementation of the graph

building/storage ($O(nm)$ for a problem with $n$ clauses and $m$ variables), a problem with 256,000 variables and 1,000,000 clauses would require $256,000 \times 1,000,000 \times 1 = 256 \times 10^9$ bits, or 32 gigabytes. Given a GPU with 32GB of memory, this would be a breaking point for the method. Of course, this is the worst case, which only occurs for a completely dense graph representation. In practice, clauses in the LEC data, for example, tend to have on average 3 or 4 variables. Since in the LCG clause nodes are only connected to the variables of which they consist, each clause node would then only have degree of 3 or 4, meaning the graph is very sparse. Thus, in practice the primary memory cost of our model scales more so according to $O(dn)$, assuming average number of variables in a clause is $d$, and assuming the implementation is adapted to leverage the sparse structure (using edge-lists instead of dense adjacency matrix, for example).

In cases where the problem is large and dense, another option might be the use of more specialized GNN methods that are specially designed to handle very large graphs. For example, by sampling from neighbourhoods or loading portions of the graph from storage.

For time-cost scaling, the primary point in the pipeline in which we suffer scaling challenges is during pre-processing. We build graphs from the text-file representation of the problems. The time-cost of this step is linear with the problem size, measured in terms of the number of clauses. Given this, time-cost is not a major factor in the scaling issue.

We will add text to clarify the scalability considerations outlined above and specify the precise nature of an "extremely large" problem that would challenge our proposed generative model.

### A.4 HardCore GNN Core Prediction Implementation Details

We implement HardCore in `DGL` using 3 Graph Convolutional Network layers combined into a hetero-GNN, where outputs of each layer are aggregated with a mean using the `hererograph` package in `DGL`. We train using 15 problems from the dataset, and we obtain training cnf-core pairs using Drat-Trim in the Core Refinement step for 200 iterations per instance. We train for 1 epoch using Binary Cross Entropy loss.

### A.5 K-SAT Random Generation

---
**Algorithm 1** Algorithm for generating 1 K-SAT Random instance.

---
$m \sim N(\mu_m, \sigma_m)$
$c \sim N(\mu_c, \sigma_c)$
$n \leftarrow \text{int}(mc)$
$\text{cnf} \leftarrow \text{randkcnf(3, m, n)}$
$\qquad\qquad\qquad\qquad\quad \triangleright$ where randkcnf(k, m, n) returns cnf m with k-var clauses from n variables.

---

Algorithm 1 shows the process by which we generated K-SAT Random instances as discussed in Section 6.1. We randomly sample hyper-parameters (number of clauses, number of variables) from a small window in order to introduce some additional variety into the dataset, and generate by randomly sampling sets of 3 variables without replacement. In our work we chose $m \sim N(400, 100)$, $c \sim N(4.4, 0.05)$.

## B   Supplementary Results

Figure 6 shows stacked histograms of the rankings for each solver, following up on the rank-1 histogram shown in Figure 5. The top row allows us to compare the ranking distribution of original LEC instances and HardCore's generations. The bottom row allows us the same for HardSATGEN. Note that the original distributions are different for HardSATGEN and HardCore because the methods were fed different quantities of data. Given HardSATGEN's cost, only 10 instances could be used for generation (to generate 50 instances), whereas HardCore is given 1445 instances and generates 5780. On inspection of the figure, we note the similarity of the original and HardCore ranking distributions. For example in HardCore, solver 1's distribution of rankings shows a very similar proportion of rank ranks 2-6, with perhaps slightly higher rank 1 (and lower rank 7) than original. In contrast, HardSATGEN shows very different distribution than the data it was given. For exmaple, we see density in rank 1 for solvers 1, 5 and 7 where there was none in the original data given to HardSATGEN. Even comparing to the true original distribution of which the top-left histogram is

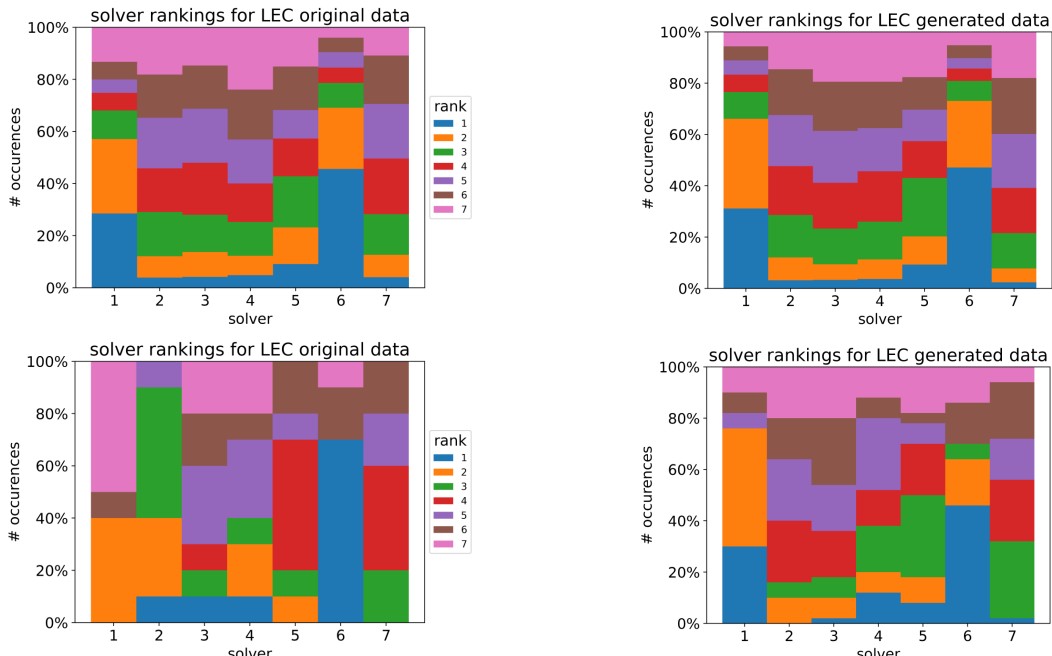

Figure 6: HardCore (top) and HardSATGEN (bottom) Comparison of Solver Ranking Histograms for Original and Generated LEC data.

representative (HardCore was given enough data to be considered a representative sample of the whole dataset), we see start differences in that solver 2 has no rank-1 density from HardSATGEN and that HardSATGEN seems to prefer solver 4 more frequently than 5 whereas the Original data favors 5 over 4 as rank-1 solver.

Table 4: GNN Core Prediction Performance

| | ↑ Core Recovery Ratio $\frac{TP}{P}$ | ↓ Core Size Discrepancy $\frac{|TP-P|}{P+N}$ | ↑Accuracy $\frac{TP+TN}{P+N}$ |
|---|---|---|---|
| Circuit-Split LEC | 0.97 | 0.05 | 0.65 |
| LEC | 0.960 | 0.009 | 0.940 |

In table 4 row-title "Circuit-Split LEC" we consider the GNN's performance on a test-set of problems originating from circuits which were not represented in the GNN's training set, as described in subsection 6.3. We note a very high recall, at the cost of acuracy and a trivial hyper-paramter tuning cost. In Table 4 row-title "LEC" we examine the classification performance of the Core Prediction GNN module. We calculate Core Recovery (which is Recall), a Size Discrepancy metric due to an observation during the design process that the Core Prediction module had a tendency to grossly over-predict and Accuracy. We find that the module performs impressively. The Core Prediction is able to identify 96% of the Core, meaning a core clause is highly unlikely to be completely missed over multiple iterations. At the same time, Accuracy is also quite high. This is important because false positives could mean the selected clause for De-Coring is in fact not a part of the core. With an accuracy of 94%, de-coring on non-core clauses will be very rare.

In Section 6.2.4 We compare the performance of two runtime prediction models: one trained on only original data and the other trained on a dataset augmented with generated instances.

To observe performance over differing levels of data availability, we conduct this experiment for several quantities of original training instances — denoted Data Size. 3 Augmentation instances are allowed per original, and augmentation is only allowed by using the original instances in the training set. Validation sets are selected from the original data only, with an 80/20 split train/validation split.

In Figure 7 we can that while there is considerable overlap with whiskers of the other methods, HardCore outperforms all other methods on all data sizes by at least one quartile of results. In

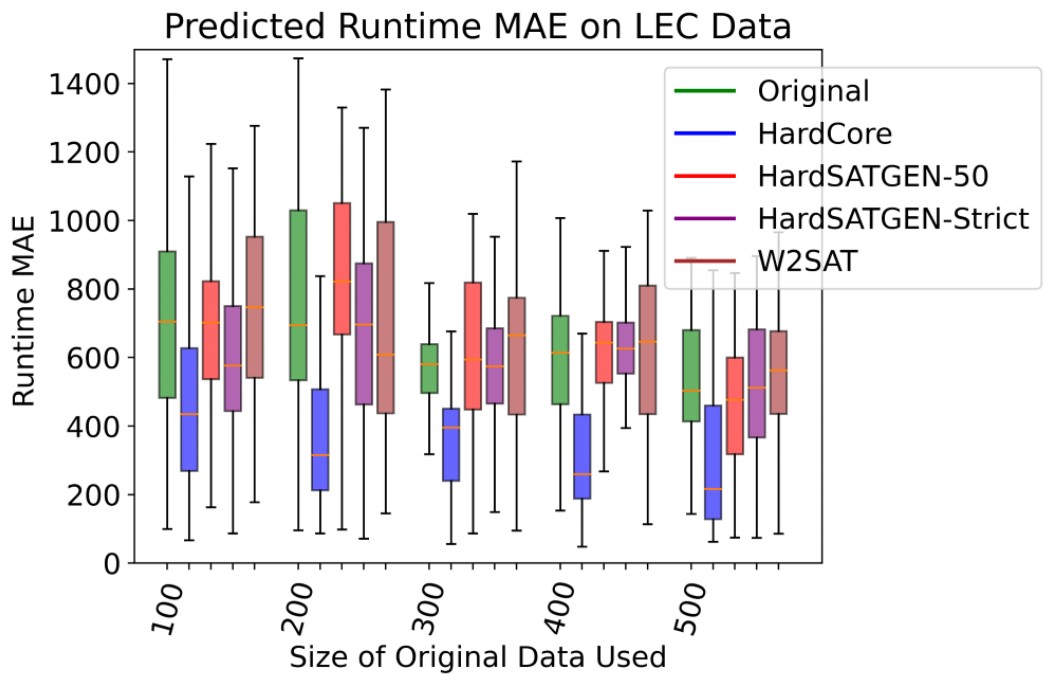

Figure 7: Mean MAE on Runtime Prediction. Boxplot-view of results presented in Table 2 for LEC data.

addition to increased prediction accuracy (lower MAE), HardCore demonstrates a tendency to reduce variance in performance, which we note by the lower whisker-to-whisker spread of the boxplots. This effect is especially notable in data-size 200, but can also be seen relative to other augmentation methods for data size 300.

In the table below we show data augmentation experimental results. Since there are only 135 problems, and our focus is on data-scarce settings, we use small datasets to train the SATZilla-based predictor. In the table, we see that for all datasets aside from the smallest one, using HardCore to generate an augmented dataset leads to a 4%-6% reduction in MAE compared to training using the original data.

Table 5: Data Augmentation experiment: MAE of Runtime Prediction averaged across 7 solvers and 15 trials. We train a runtime prediction model according to the experimental setting in 6.4.2 of the paper. Columns in the table indicate the number of original problems used in the training set (we generate 4 times per original problem in the training set). Results in the "HardCore" row are MAE for a runtime prediction model trained on HardCore-augmented data, whereas "Original" indicates un-augmented performance.

| | Tseitin Dataset Size | | | | |
|---|---|---|---|---|---|
| Training Data | 10 | 20 | 30 | 40 | 50 |
| HardCore-Augmented | 3618.9 | **3410.0** | **3311.4** | **3417.7** | **3419.4** |
| Original (Un-Augmented) | **3369.6** | 3581.9 | 3576.1 | 3544.7 | 3608.5 |

Table 6: Data Augmentation experiment: MAE of Runtime Prediction averaged across 7 solvers and 15 trials. We train a runtime prediction model according to the experimental setting in 6.4.2 of the paper. Columns in the table indicate the number of original problems used in the training set (we generate 4 times per original problem in the training set). Results in the "HardCore" row are MAE for a runtime prediction model trained on HardCore-augmented data, whereas "Original" indicates un-augmented performance.

| | FDMUS Dataset Size | | | | |
|---|---|---|---|---|---|
| Training Data | 100 | 200 | 300 | 400 | 500 |
| HardCore-Augmented | **0.220** | **0.197** | **0.162** | **0.142** | **0.142** |
| Original (Un-Augmented) | 0.246 | 0.213 | 0.184 | 0.173 | 0.145 |

