# OpenReview forum: "HardCore Generation: Generating Hard UNSAT Problems for Data Augmentation"
_NeurIPS.cc/2024/Conference — NeurIPS 2024 poster_

### Official Review · Reviewer_1y8R · 2024-06-20

**Soundness:** 4
**Presentation:** 4
**Contribution:** 3
**Rating:** 7
**Confidence:** 5

**Summary:**

This paper presents a procedure for synthesizing hard UNSAT formulas from a given distribution of UNSAT formulas. Concretely, a formula is generated by 1) extracting a core from a seed instance; 2) adding random new clauses; and 3) refining the formula to become harder. In step 3), a GNN is trained to predict UNSAT core of a given formula and the predicted core is relaxed, potentially yielding a larger core. On two problem distributions, LEC Internal and random K-SAT, the proposed method can generate hard instances faster than previous approaches, and the generated instances exhibit similar hardness distribution.

**Strengths:**

- The proposed method strikes a good balance between the hardness of the generated instances and the generation time. It can generate instances much harder than W2SAT, while in a much faster manner than HardSATGEN.
- The combination of the two ideas, iteratively relaxing core to generate hard instances and learning to predict cores, is quite clever and elegant.
- The fact that the solver performance on the synthetic instances resembles that on the original instances is a good indication that the generated instances are similar in some way to the original instances.

**Weaknesses:**

- It seems that the proposed core refinement technique can be viewed as a post-processing step that can be applied to any initial formulas (e.g., ones generated by W2SAT). Have the authors considered building on top of previous DL-generated formulas? Compared with formulas generated by randomly adding clauses to an original UNSAT core, conceptually, formulas generated by a method like W2SAT, which are trained to embed a formula distribution, seem to be a more justifiable starting point.
- The method can still only handle relatively small formulas.
- Given that the LEC benchmarks are proprietary, only random K-SAT will be made publicly available. However, one could have considered using dataset such as SATLIB like in previous work (e.g., SATGEN, G2SAT, W2SAT).

Minor:
- x axis labels are missing in Figure 5.
- Line 353: "syntehtic" -> synthetic

**Questions:**

1. We have previously argued that random data differs from real data in important ways that make it unsuitable for machine learning applied to real problems. Could you elaborate?

**Limitations:**

The authors adequately addressed the limitations. No potential negative societal impact.

---

> ### Author Rebuttal · Authors · 2024-08-07
>
> We deeply appreciate the reviewer's generous acknowledgement of the
> elegance and performance of our method. In addition, the reviewer's
> comments about scalability and publicly available data have led us to
> make significant additions to the depth of our work's experimental
> setting and analysis. Finally, the reviewer has provided an insightful
> future direction regarding the generation mechanism used in our method.
>
> ## Weaknesses
>
> 1.  *Core refinement can be viewed as post-processing applied to any
>     generator. Have authors considered building on top of previous
>     DL-generators? Starting with methods like W2SAT which are trained to
>     embed formula distributions seems more justifiable than starting
>     with a random generator.*\
>     Viewing core refinement here as a post-processing step applicable to
>     any generator is valid. During the design of the proposed method, we
>     did consider building on top of previous DL-generators. However, a
>     random generator designed to emulate industrial problems was
>     selected for its simplicity, minimal cost, and the convenience of
>     not having to train it. Indeed, perhaps using a DL generator may
>     improve the downstream performance. Investigating such possibilities
>     could prove insightful and is a valuable direction for future work.
>
> 2.  *The method can still only handle small formulas*\
>     While scalability is certainly a concern as discussed in the
>     Limitations section of the text, the breaking point of our method
>     for a 32GB GPU, for example, would be in worst-case a graph with
>     hundreds of thousands of variables and clauses. This is a problem
>     size beyond many real-world applications. In the LEC data, no
>     problem exceeds 100,000 clauses. This limitation was primarily
>     expressed in consideration of the SAT Competition data, in which
>     problems (almost always randomly generated problems) can sometimes
>     reach a million clauses or more. Our scalability limitation, as
>     discussed in the general response, is primarily related to memory,
>     and the storage of the adjacency matrix of the graph. A potential
>     direction for future work is to investigate the application of
>     existing high-efficiency GNN methods that are capable of addressing
>     graphs of tens-of- millions of nodes.
>
> 3.  *Only K-SAT will be made publicly available. Why not use dataset
>     such as SATLIB like in previous work?*\
>     We decided against using SATLIB as the great majority of its
>     problems are either randomly generated or not UNSAT (our method is
>     limited to UNSAT problems). Additionally, similarly to the SAT
>     competition benchmarks, the data is composed of problems from
>     several highly distinct sources, which is undesirable for
>     machine-learning applications. Despite this, we have included in the
>     global rebuttal and single-page pdf results on data from the SAT
>     Competition, showing that our method is able to generate hard
>     problems and provide data augmentation for SAT Competition data. We
>     will make the code for our SAT Competition experiments publicly
>     available.
>
> ## Questions
>
> 1.  *It has been argued that random data differs from real data in
>     important ways that make it unsuitable for machine learning applied
>     to real problems. Could you elaborate?*\
>     The clearest demonstration of the difference between real data and
>     random data, in our view, is the notable difference in solver
>     performance. It is generally accepted in the literature that some
>     solvers are random-specialized and others are industry-specialized.
>     For example, the experimental results in \[1\] are heavily based on
>     this. One can also see such specialization by noting that the best
>     solver on the SAT Competition random tracks is not the same as the
>     best solver on the SAT Competition main track. In fact, the
>     existence of separate tracks itself indicates an acknowledged
>     distinction between the types of data.
>
>     Given this distinction between the behavior of random and industrial
>     data, training an industrial-facing model on random data leads to an
>     out-of-distribution problem. Inference must be performed for
>     problems that are very different from the training data. It would be
>     equivalent to augmenting a real-world image dataset with random
>     patterns. In most cases the model will perform poorly. For example,
>     SAT-solver selection is a machine-learning task for SAT in which,
>     given a problem, we aim to rapidly choose the solver which is likely
>     to solve the problem the fastest. This is a task which greatly
>     benefits industrial settings in which thousands of SAT problems must
>     be solved each day and computation and time costs must be minimized.
>     Training such a selection model on random data would result in the
>     model learning a wholly different runtime distribution than that of
>     the industrial data. A similar example is low-cost benchmarking, in
>     which a model is trained to predict the performance of a solver over
>     a benchmark. This task is of interest to solver-designers, as it can
>     be much cheaper than running a design-iteration of a new solver on
>     the whole benchmark data. If random data is used to learn to
>     benchmark a solver on industrial problems, the predictor would
>     likely predict benchmarks as if the industrial data were in fact
>     random. A third example is the task of hyper-parameter tuning of
>     solvers, as presented in the HardSATGEN paper as a down-stream task.
>     If one were to use random data to tune the hyper-parameters, the
>     solvers would likely be poorly tuned for industrial data, whose
>     hardness distributions are very different to random data.
>
>     [1]J. Giraldez-Cru and J. Levy, "Generating sat instances with
>     community structure," in Artificial Intelligence, 2016, p. 119--134.

---

> > ### Comment · Reviewer_1y8R · 2024-08-12
> >
> > I thank the authors for their clarifications and new results. I would like to keep my score.

---

### Official Review · Reviewer_vCMn · 2024-07-14

**Soundness:** 2
**Presentation:** 2
**Contribution:** 2
**Rating:** 4
**Confidence:** 5

**Summary:**

This paper proposes a novel method for generating hard UNSAT problems. The method targets the "core identification" problem and iteratively performs refinement using a GNN-based detection procedure, which preserves the key aspects of the original instances that impact solver runtimes. The experimental results show that the method can generate instances with a similar solver runtime distribution as the original instances in a short time.

**Strengths:**

This paper introduces a novel method to generate hard UNSAT problems in a reasonable time frame, which alleviates the data scarcity problem in the SAT solving domain and improves the performance of deep learning methods in this area. Compared to previous methods, this method can generate SAT problems that are more similar to the original instances.

**Weaknesses:**

1. There is not enough support for the correlation between hardness and core size. For example, in the random 3-SAT problem, the UNSAT core can constitute a significant portion of total instances, even more than 80%, yet modern solvers can easily address these instances.
2. Lack of details about model training. There are no explanations on how to prepare the training dataset.
3. The experimental setting is not convincing. Authors only perform the proposed hard case generation approach on LEC and random k-SAT problems, however, there are no details about how to construct these two datasets. Moreover, it is essential to show results on other datasets to showcase the generalization ability of the proposed method, such as the SAT competition benchmarks.
4. There are some logical problems in the paper’s writing. In the “Core Refinement” part, the paper states that the addition of random new clauses is likely to create a trivial core. However, I find that in the core refinement pipeline, there is no addition of new clause. Instead, new literal is added. The writing here is kind of messy.

**Questions:**

1. Please provide justifications on the correlation between solving hardness and UNSAT core size.
2. Even though considering the minimal UNSAT core, one instance may contain multiple cores. How to handle this one-to-many mapping problem? This problem is significant as it is directly related to the supervision.
3. The random k-SAT and LEC cases have a very strong bias compared to the real instances. The authors should also prove the generalization ability of their proposed approach on more diverse datasets, such as SAT competition benchmarks.
4. In Section 6.2.3, you mention that you use a subset of the training set as the evaluation dataset. This is not the right.
5. When you perform the core refinement step, can the problems generated in the middle be used? How does n, the number of refinement steps, affect the characteristics and performance of the generated problems?

**Limitations:**

Yes, the authors have stated the limitations of their work.

---

> ### Author Rebuttal · Authors · 2024-08-07
>
> We thank the reviewer for their careful and critical comments. These
> comments have prompted several new explorations, including an
> examination of the progression of hardness during refinement, and the
> addition of a new dataset to the experimental setting. These improve
> both the strength of our results and the depth of our analysis of the
> method.
>
> ## Weaknesses
>
> 1.  *Correlation between hardness and core size is not sufficiently
> supported.*
>
> We do not claim in the paper that there is a correlation between the
> core size and the hardness. For our proposed
> technique to work, our key assumption is that if a problem is easy
> then core prediction can identify the easiest core, and then by
> removing that core the problem is made harder. The experimental results
> demonstrate that we generate challenging problem instances,
> providing support that our assumption is correct. We agree with the
> reviewer that there is value in providing more direct evidence.
>
> First, we evaluate the hardness of the original problems in the LEC dataset. Note that this is a
> real-world dataset derived during industrial circuit design. Figure
> 3 (left) in the attached pdf shows that there is a general trend of
> the hardness increasing as the core size increases, up to a
> threshold of 4000-5000 clauses. This trend is observed for absolute
> core size rather than the percentage of clauses in the core; the
> reviewer's observation is correct that an 80% core can be easy.
>
> Although we observe this correlation, it
> is not essential for our method. Much more important are the results
> shown in Figure 3 (right) in the pdf, where we show how the hardness
> changes as a result of refinement. The figure shows
> boxplots of the percentage change in hardness for different bins of
> initial hardness. For easy problems we increase the
> hardness sometimes by a factor of over 200, indicating successful
> hardening of hardness-collapsed problems.
>
> 2.  *Model training details missing.*
>
> Details on how the training data is prepared for the core prediction
> model (In particular, how supervision labels are generated) is
> provided at l. 233.
>
> There are three **separate** groupings of the dataset: (i) Core
> Prediction training data, (ii) generation seeding data, and (iii)
> the remaining data. This split is chosen randomly. Core
> Prediction training data can be small (we used 15 problems),
> because we use each problem as a seed instance 5 times for generation
> followed by core-refinement with a traditional core detector. Saving problem-core pairs at each step,
> we obtain 15,000 training pairs for the
> core-predictor model. The seeding data are used to
> seed HardCore once the core predictor is trained in order to obtain
> generations to evaluate. These generations are then compared against
> the seed data for runtime similarity. Finally, these generations
> (and their seeds) are used to train a runtime-predictor model,
> which is evaluated on the remaining un-used data.
>
> The core prediction model is trained with a binary cross-entropy
> loss using the prepared data.
> For more details on the training procedure, please consider the
> code provided in the supplementary material. We will add a detailed
> description of this training process in an appendix.
>
> 3.  *only 2 datasets, no SAT Competition benchmarks. *
>
> Please see the global rebuttal, in which we present new results on
> SAT Competition data.
>
> 4.  *Construction details of the used datasets are not provided.*
>
> For details on how the K-SAT dataset was generated, please see
> Appendix section A.4. The problems from the LEC dataset are
> SAT problems which were solved during the design of real circuits,
> and were saved for future use in analysis and research.
>
> 5.  *Unclear writing: in \"Core Refinement\", the paper states that
> adding new clauses is likely to create trivial cores, but clauses
> are not added during core refinement.*\
>
> In the text, the correction will be at line 176: "The addition of
> random clauses during **generation** is very likely to [...]".
> Clauses are not added during refinement, the reviewer is correct.
>
> ## Questions
>
> 1.  *Justify core-size and hardness correlation*.\
>
> See Weakness 1.
>
> 2.  *One instance may have multiple cores. How to handle this one-to-many mapping problem.*\
>
> The goal of the estimator is not to only the easiest core.
> The first core to be output by the detection algorithm is considered the easiest
> (finding a core proves the problem UNSAT, solving the problem).
> We use the outputs of the core detector as labels for
> supervision, and find that the GNN is able to identify the easiest
> core based on performance results shown in Table 4 of Appendix B of
> the paper. Once that core is de-cored, the other core may now be the
> easiest, and it will be detected in the next refinement step.
>
> 3.  *k-SAT and LEC cases have very strong bias compared to real
> instances.*\
>
> We believe the LEC cases we use in the paper must be considered as
> real instances as they come from real industry.
> Logic Equivalence Checking (LEC) is a critical step in
> logic synthesis for circuit design. The LEC data we use in the paper came from the
> design pipeline of 29 industrial circuits by a prominent circuit
> design company.
> 4.  *The authors should show generalization ability on the SAT
> competition data.*\
>
> See weakness 3.
>
> 5.  *In section 6.2.3 it is mentioned that a subset of training data is
> used as evaluation data. This is not right.*\
>
> We do not evaluate with training data at any point in the
> experimental method. We have been unable to locate any reference to
> this in Section 6.2.3. If you could kindly quote the line (or
> paragraph) in which this was conveyed, we would be very happy to
> clarify the issue.
>
> 6.  *Can problems from halfway through core refinement be used? How does
> the number of refinement steps affect performance?*\
>
> Yes, although they will likely be easier. Problems usually get harder in
> a smooth progression during refinement, as shown in Figure 1 (left).

---

> > ### Comment · Reviewer_vCMn · 2024-08-11
> >
> > Thank you for the responses, which clarify most concerns. I'll raise my score accordingly. However, I still have questions regarding Weaknesses 1 and 3.
> > 1. I understand that the proposed approach adds literals to clauses in the UNSAT core rather than creating additional clauses. However, there seems to be a conflict, as mentioned in Figure 2: “As steps (1) and (2) are repeated, the core gradually becomes larger, raising the hardness of the generated instance.” This sentence should be further clarified to avoid misunderstandings.
> > 2. I noticed the global rebuttal results that demonstrate generalization ability. However, the selected "Tseitin" family likely shares a similar distribution with LEC problems, since LEC data is also converted into CNF using the Tseitin Transformation. Could the authors clarify this assumption? I suggest the authors test the proposed approaches on non-circuit-based families and novel cases in SAT benchmarks, such as cryptography and scheduling.

---

> > > ### Author Response · Authors · 2024-08-11
> > >
> > > We express our gratitude to the reviewer for acknowledging our response.
> > >
> > > 1.  *I understand that the proposed approach adds literals to clauses in
> > >     the UNSAT core rather than creating additional clauses. However,
> > >     there seems to be a conflict, as mentioned in Figure 2: "As
> > >     steps (1) and (2) are repeated, the core gradually becomes larger,
> > >     raising the hardness of the generated instance." This sentence
> > >     should be further clarified to avoid misunderstandings.*\
> > >     \
> > >     We agree that the sentence needs to be edited and we will revise it.
> > >     We will amend the text to clarify that the core we consider at each
> > >     step is specifically the easiest core (because traditional core
> > >     detection effectively solves the SAT problem). We also intend to
> > >     replace references to the "enlarging" of cores with their
> > >     "hardening" in order to emphasize our crucial assumption as
> > >     discussed previously. Thus, this phrase would read: "As steps (1)
> > >     and (2) are repeated, the easiest core of the problem is gradually
> > >     refined, raising the hardness of the generated instances."
> > >
> > > 2.  *I noticed the global rebuttal results that demonstrate
> > >     generalization ability. However, the selected \"Tseitin\" family
> > >     likely shares a similar distribution with LEC problems, since LEC
> > >     data is also converted into CNF using the Tseitin Transformation.
> > >     Could the authors clarify this assumption? I suggest the authors
> > >     test the proposed approaches on non-circuit-based families and novel
> > >     cases in SAT benchmarks, such as cryptography and scheduling.*\
> > >     While it is true that the LEC and Tseitin data share an element of
> > >     origin, we note that the Tseitin data is made up of problems
> > >     generated in many different but random ways. For example, the
> > >     data-points contributed by Elfers in the 2016 Hand-Crafted track
> > >     \[1\] are generated by providing grid-graphs of varying dimensions
> > >     to the Tseitin transformation. These grid-graphs are very different
> > >     from the circuit-graphs seen during LEC. In consequence we see very
> > >     different runtime-distributions. Comparing Figure 2 in the rebuttal
> > >     pdf to Figure 6 in Appendix B of the paper and Figure 4 in the text,
> > >     we note significant differences. Most notably, Solvers 1 and 6
> > >     (which we generally have found to be strong for industrial data and
> > >     weak for random data, are much slower for the Tseitin data).
> > >     Therefore, we do not believe the Tseitin data shares a similar
> > >     distribution to the LEC data as the reviewer has suggested.
> > >
> > >     Additionally, while other data such as Cryptography and Scheduling
> > >     were considered, Tseitin was the family with the most UNSAT problems
> > >     available in the benchmarks. Since more data enable us to provide
> > >     more complete distributional experimentation as well as more results
> > >     on the data augmentation task, we chose the largest dataset: the
> > >     Tseitin.
> > >
> > >     With that said, we have begun experiments on a dataset of 100
> > >     scheduling problems taken from the SAT Competition data using the
> > >     filter "`family=scheduling and result=unsat`". Of the two families
> > >     suggested by the reviewer, scheduling had more problems, and so we
> > >     chose it over cryptography. If time permits for completion before
> > >     the end of the discussion period we will share results. Otherwise,
> > >     we will report the fully processed results in the appendix of the
> > >     revised paper.
> > >
> > >     \[1\] *Proceedings of SAT Competition 2016: Solver and Benchmark
> > >     Descriptions*, volume B-2016-1 of Department of Computer Science
> > >     Series of Publications B, University of Helsinki 2016. ISBN
> > >     978-951-51-2345-9.

---

### Official Review · Reviewer_kFMW · 2024-07-14

**Soundness:** 3
**Presentation:** 3
**Contribution:** 3
**Rating:** 6
**Confidence:** 5

**Summary:**

This paper introduces HardCore, a novel method for efficiently generating hard Unsatisfiable (UNSAT) Boolean Satisfiability (SAT) problems, addressing the critical challenge of data scarcity. The approach combines a Graph Neural Network (GNN) for rapid core prediction with an iterative core refinement process, enabling the generation of thousands of hard instances in minutes or hours while preserving problem difficulty. HardCore outperforms existing methods in terms of generation speed and hardness preservation, as demonstrated through experiments on both Logic Equivalence Checking (LEC) data and K-SAT Random data. Despite limitations such as applicability only to UNSAT problems and potential scalability issues with extremely large instances, HardCore represents an advancement in SAT problem generation.

**Strengths:**

* The work provides a meaningful contribution to the SAT community by introducing a novel method that generates SAT instances within a reasonable timeframe while preserving the hardness of the original problems. The work has potential applications in various industrial settings, though the scalability limit of the approach might pose a challenge.
   * The authors conduct extensive experiments, comparing their method against multiple baselines and evaluating various aspects such as hardness preservation, generation speed, and similarity to original distributions. The paper compares against relevant and recent baselines (HardSATGEN, W2SAT, G2MILP), providing a comprehensive view of how HardCore performs relative to the state-of-the-art.
   * Innovative core refinement process: The iterative core refinement process (Figure 2) is a clever approach to gradually increasing problem hardness while avoiding the creation of trivial cores.

**Weaknesses:**

* The experiments primarily focus on two datasets (LEC Internal and K-SAT Random), with one being proprietary. This somewhat limits the generalizability of the results.
   * While the paper focuses on runtime distributions and solver rankings, it lacks a deeper analysis of the structural properties of the generated instances (e.g., clause-to-variable ratios, community structure) compared to the original ones.
   * The core refinement process (Section 5.1) involves adding a single literal to break easy cores. The paper doesn't explore more sophisticated strategies or justify why this simple approach is sufficient.
   * The paper mentions struggles with "extremely large SAT problems" but doesn't clearly define what constitutes "extremely large" or provide empirical evidence of where the method breaks down.

**Questions:**

How would one extend your technique to more structured classes of formulas?

**Limitations:**

It would really strengthen the paper if the authors were to study the structural properties (e.g., clause-variable ratio, tree width, hierarchical community structure,...) of the generated instances and identify why they are hard for modern SAT solvers.

---

> ### Author Rebuttal · Authors · 2024-08-07
>
> We wish to express our gratitude to the reviewer for their
> acknowledgement of our extensive experimentation and innovation. The
> reviewer's comments have guided us to explore meaningful improvements to
> the presented work: complete results on an additional public dataset, a
> deeper analysis of scaling considerations and comparison of SAT-problem
> statistics. The reviewer has also illuminated multiple promising avenues
> for future work.
>
> ## Weaknesses
>
> 1.  *The experiments focus on only 2 datasets, limiting result
> generalizability*
>
> Please see the global rebuttal for the
> presentation of results on a new dataset in order to address this
> issue. We introduce some data from the SAT Competition database and
> present results which show that HardCore is able to generate
> problems which resemble the SAT Competition (SC) data in hardness.
> Results also show that HardCore is able to provide data augmentation
> which improves a runtime-prediction model's performance by 4-6%.
>
> 2.  *Lacks "deeper analysis" of structural properties
> (clause-to-variable ratios, community structure)*
>
> We chose not to report such results in the paper because such statistics have neither been claimed nor shown
> to have causal connections to hardness or industrial structure, only correlation.
>
> Despite this, we agree with the reviewer that the paper would
> benefit by a presentation of the structural properties of the
> instances generated by HardCore. In the table below, we compare the
> statistics of the generated instances to those of the original
> problems, replicating the experiment that was peformed in the
> HardSATGEN paper. We report the number of variables and clauses,
> clustering on the variable-instance graph, and modularity on four
> graph-representations. Note that both HardCore and HardSATGEN only
> add a single auxiliary variable (to eliminate easy cores), so the
> number of variables is a close match. HardCore maintains the number
> of clauses by setting the number of generated clauses (an easily-set
> parameter of the generation algorithm) accordingly. For other
> statistics, HardCore achieves values that are relatively close to
> those of the original instances, with a slightly reduced modularity
> in the VIG and LIG representations, compared to HardSATGEN. This is
> likely because HardSATGEN monitors VIG communities while HardCore
> does not consider incidence representations.
>
> |              | HardSATGEN | HardCore  | original |
> |--------------|------------|-----------|----------|
> | num. vars    | **933.8    | **933.8** | 932.8    |
> | num. clauses | 3395.16    | **3400.2  | 3400.2   |
> | VIG clust.   | 0.38       | **0.39    | 0.39     |
> | mod. VIG     | **0.74**   | 0.56      | 0.74     |
> | mod. LIG     | **0.74 **  | 0.63      | 0.75     |
> | mod. VCG     | **0.81     | 0.78      | 0.81     |
> | mod. LCG     | **0.64**   | **0.64    | 0.67     |
>
>
> It is our view that hardness distributions --- whether per-solver,
> ranking, multi-solver --- are considerably more important than the
> measured structural statistics presented in previous works. Other
> generators, despite being capable of preserving structural
> statistics reasonably well, generate trivial problems. These clearly
> differ in critical ways from the seed instances from the perspective
> of the solvers. For example, W2SAT reports modularity very similar
> to the original problems' modularity and yet fails to generate hard
> instances. Since HardCore is directly shown to preserve hardness
> behavior well, we did not strive for similar modularity.
>
> 3.  *De-coring is done by adding a single literal to break cores. More
> sophisticated strategies are not explored. Why is this approach
> sufficient?*
>
> Exploring more sophisticated de-coring strategies could
> be an interesting direction for future work, especially for
> application to more structured formula families. We
> chose to use the established de-coring paradigm from the HardSATGEN
> method, considering it to be interpretable due to its
> simplicity. More complicated strategies, which might add or remove
> existing variables from the clauses (instead of adding new ones),
> could introduce unforeseen conflicts and cores which are easier than the current de-coring target. The
> current strategy guarantees the removal of the current core, without
> constructing a new one, which is desirable. Since the approach
> proved experimentally effective, we did not explore other
> strategies.\
>
> 4.  *The paper mentions struggles with extremely large problems. Define
> what constitutes "extremely large" or provide empirical evidence of
> where it breaks down*
>
> Please see our discussion of scalability in
> the global rebuttal, in which we describe the memory-scaling in
> worst-case dense problems and more realistic sparse problems and in
> which we discuss worst-case break-down given our hardware.
>
> ## Questions
>
> 1.  *How would one extend the work to more structured classes of
> formulas?*
>
> For highly structured problems, the generation mechanism would have
> to be designed such that a formula adheres to the required
> structure.
>
> Structured problems such as the pigeonhole problem and
> graph-coloring are often easy to generate. For example, a graph
> coloring problem can be generated by generating a graph in any
> random way, and then applying the coloring problem. A pigeonhole
> problem can be generated simply by parameters for the
> problem.
>
> Core Prediction itself would likely remain un-changed. In fact, if
> the data are focused on one structured
> class of formulas, we would expect high performance from the
> prediction model.
>
> Special care would be required to ensure that the
> de-coring operation does not corrupt the required structure of the
> problem. By adding a new variable to
> de-coring target clauses, we may change structure of the problem
> in such a way that it no longer conforms to the strict
> structure of the family. De-coring operations would therefore have
> to be specially designed for each family type.
>
> This represents an interesting challenge and is an intriguing
> direction for future work.

---

### Official Review · Reviewer_RF4y · 2024-07-21

**Soundness:** 4
**Presentation:** 4
**Contribution:** 3
**Rating:** 7
**Confidence:** 4

**Summary:**

This paper addresses the scarcity of practical data (industrial satisfiability problem instances) for training deep learning methods for SAT solving. The existing data augmentation methods suffer from either the limited scalability or the collapsed hardness of the generated instances. Therefore, the authors introduced a fast augmentation method while preserving the original data's hardness. The primary technical contribution is a fast UNSAT core detection procedure using graph neural networks to monitor the hardness of UNSAT formulae during data augmentation. The empirical evaluation confirmed the new method’s fast generation speed and the preserved hardness of the generated instances. In the application of solver runtime prediction, the augmented data led to a 20-50 percent reduction in mean absolute error.

**Strengths:**

The proposed method achieved the best of both worlds. It has a similar speed to a fast generator while preserving a similar hardness of generated instances to a high-quality but slow generator.

The proposed idea is simple and easy to follow, yet effective. The paper is well-written.

The authors performed extensive evaluations to demonstrate the preserved hardness of the generated instances in various aspects, such as the preserved average hardness, the hardness distribution over instances, the runtime distribution per solver, and the best-solver distribution over instances.

The authors illustrated the effectiveness of their method in a practical application (solver runtime prediction). The augmented data successfully reduced the mean absolute error by 20-50 percent.

**Weaknesses:**

The authors don’t present how well the core predictor generalizes to other datasets. We may need to re-train the core predictor when applied to a new family of instances if it doesn’t generalize well, which would introduce extra overhead not presented in the current evaluation.

The technique to eliminate an easy core described in lines 176-191 has been proposed in [1]. The authors should cite the work and move the content to related work.

[1] Yang Li, Xinyan Chen, Wenxuan Guo, Xijun Li, Junhua Huang, Hui-Ling Zhen, Mingxuan Yuan, and Junchi Yan. 2023. HardSATGEN: Understanding the Difficulty of Hard SAT Formula Generation and A Strong Structure-Hardness-Aware Baseline. In Proc. of ACM SIGKDD Conf. Knowledge Discovery and Data Mining

In line 190, the satisfying solution should be (A=0, B=0, C=1).

The axis label is missing in Fig. 5.

**Questions:**

1. How much time did you spend on collecting data and training the core predictor? If the core predictor doesn’t generalize well to a new family of instances, is the re-train time going to be a huge overhead of your method?
2. How many instances are generated from a single seed instance by HardCore and HardSATGEN? If you could use one seed instance to generate multiple instances, why can’t you keep the same number of seed instances for HardCore and HardSATGEN in line 304?
3. Why do you use the extremely small data size (e.g., 10 and 100) in Table 2? It should be considerably insufficient, even after the three-time augmentation, to train a fair runtime predictor.

**Limitations:**

The limitations have been well addressed. The authors acknowledged that the method can only be applied to UNSAT instance generation and is not suitable for the SAT case. They also mentioned that the results are limited to the current datasets and their method can’t scale to large SAT problems with millions of clauses.

---

> ### Author Rebuttal · Authors · 2024-08-07
>
> We would like to thank the reviewer for their insightful and
> constructive review. In following the reviewers comments we have
> uncovered new, valuable experiments (Circuit-Split LEC) as well as
> short-comings in the description of certain details of the experimental
> setting.
>
> ## Weaknesses
>
> 1.  *GNN generalization to other datasets not shown. If we have to
>     re-train for new data, this consists of overhead.*\
>
>     In order to measure GNN generalization to new data without
>     re-training, we create a new split of the LEC data. Each problem in
>     the LEC data can be traced back to one of 29 circuits. By randomly
>     splitting circuits into training circuits and test circuits (and
>     then building training and evaluation sets with their respective
>     problems), we can measure generalizability. Note that we would not
>     expect the model to generalize to problems derived from a completely
>     different application domain (although fine-tuning a previously
>     model in a domain adaptation strategy might be interesting to
>     explore).
>
>     In Table 2 we report the GNN performance on this experiment. In the
>     paper we discussed that Core recovery is the priority, because if we
>     falsely classify true-positives then we may be un-able to de-core
>     the current core (since the necessary clause may be un-detected),
>     whereas if we mis-classify true-negatives then we will simply
>     de-core a non-core clause. Given enough iterations of
>     core-refinement, a true-positive clause will eventually be selected
>     for de-coring (since the clause for de-coring is randomly selected
>     from among the detected clauses). With this in mind, the threshold
>     hyper-parameter which is used on the sigmoid outputs at model
>     readout becomes a useful parameter in cases where classification
>     performance is weakened: we can boost Core Recovery (recall) by
>     lowering the threshold. Tuning this threshold is very low-cost:
>     testing a thousand problems takes 500 seconds on a GPU. We find that
>     by testing values $[0.1, 0.3, 0.5, 0.7, 0.9]$ --- which takes 25
>     minutes --- we can tune the threshold to provide similar recall to
>     the in-distribution model.
>
> |                   | ↑ Core Recovery Ratio (Recall) | ↓ Core Size Discrepancy (TP-P)/(P+N) | ↑Accuracy  |
> |-------------------|--------------------------------|--------------------------------------|------------|
> | Circuit-Split LEC | 0.97                           | 0.05                                 | 0.65       |
> | LEC               | 0.960                          | 0.009                                | 0.940      |
>
>
> 2.  *The de-coring method of adding a literal to a clause was put
>     forward by the HardSATGEN paper. HardSATGEN should be cited and this
>     should be moved to related work.*\
>     Thank you for this comment. We will modify the paper to acknowledge
>     this and cite HardSATGEN accordingly. We did not intend to imply
>     that this was a contribution of our work.
>
> 3.  *Correct Satisfying solution at l. 190* Thank you, we will make this
>     correction.
>
> 4.  *Fig. 5 axis label missing* Thank you, we will correct this in the
>     final version.
>
> ## Questions
>
> 1.  *How long does it take to collect training data and train the core
>     predictor for new family of instances?*\
>     Data collection requires the execution of iterative core refinement
>     using a traditional core detection algorithm. This process is indeed
>     costly. With 200x parallelization, which is easily achievable using
>     a moderately-sized server, data collection for training the GNN
>     predictor takes approximately 26 hours. The training itself of the
>     GNN is relatively cheap computationally, taking only 135 minutes.
>     While there are thus training costs, we stress that the generation
>     is very low-cost, which allows us to amortize training cost over
>     many generations. For example, HardCore can generate 20,000 LEC
>     instances in 24 hours, once trained. This means that the total time
>     required to generate those 20,000 instances, including
>     training-time, is 24+26+2.25 hours, which amounts to 9 seconds per
>     instance. This is very low compared to HardSATGEN's generation cost
>     for LEC of 6441 seconds per instance, even without including its
>     training time-cost.
>
> 2.  *How many instances are generated per seed?*\
>     We generate 5 instances per seed using each method. We will add this
>     information to the experimental setting in the text.
>
> 3.  *If we can use a seed more than once, why can't we obtain many
>     HardSATGEN generations?*\
>     With HardSATGEN, the primary bottleneck is the slow iterative
>     core-detection step during generation. Given that this must be
>     performed for each generation (and not only for each seed),
>     generating 100 instances from only 1 seed (100 instances per seed)
>     with HardSATGEN has the same cost as generating 100 instances from
>     100 seeds (1 instance per seed). HardSATGEN's generations were
>     limited during our experiments by time budgets for generation, not a
>     scarcity of seeds (we have as many seeds for HardSATGEN as we did
>     for HardCore and the other baselines).
>
> 4.  *Why do we use such small data in Table 2 (runtime-prediction
>     experiment) It should be insufficient to train a fair runtime
>     predictor?*\
>     The motivation for our work is that SAT data is often scarce. This
>     is especially the case for public datasets. Therefore in this
>     downstream task, our goal was to model a realistic case, where one
>     is provided with only a small number of examples in the dataset for
>     training the predictor.

---

> > ### Comment · Reviewer_RF4y · 2024-08-13
> >
> > Thank you for your response!

---

### Author Rebuttal · Authors · 2024-08-07

We thank the reviewers for the insightful and thoughtful reviews. All
reviewers have stated that the proposed method is novel and stands as a
meaningful and innovative contribution to the field. Several reviewers
agreed that the experimentation was extensive and demonstrative of
improvements over existing work. The primary concerns expressed in the
reviews were (1) scalability and cost overhead; (2) result
generalizability to other datasets; and (3) certain details which
required clarification. We have addressed these concerns in the response
by (1) providing a more detailed cost analysis for the pre-processing of
data; (2) presenting new results on a public dataset; and (3) providing
additional clarifications, details and analysis where required.

\(1\) Scalability: Memory limitations are the primary challenge for our
proposed method, due to the need to store the graph. For our
experimental hardware (32GB GPU) and our implementation of the graph
building/storage ($O(nm)$ for a problem with $n$ clauses and $m$
variables), a problem with 256,000 variables and 1,000,000 clauses would
require $256,000 \times 1,000,000 \times 1 = 256 \times 10^9$ bits, or
32 gigabytes. Given a GPU with 32GB of memory, this would be a breaking
point for the method. Of course this is the worst case, which only
occurs for a completely dense graph representation. In practice, clauses
in the LEC data, for example, tend to have on average 3 or 4 variables.
Since in the LCG clause nodes are only connected to the variables of
which they consist, each clause node would then only have degree of 3 or
4, meaning the graph is very sparse. Thus, in practice the primary
memory cost of our model scales moreso according to $O(dn)$, assuming
average number of variables in a clause is $d$, and assuming the
implementation is adapted to leverage the sparse structure (using
edge-lists instead of dense adjacency matrix, for example).

In cases where the problem is large and dense, another option might be
the use of more specialized GNN methods that are specially designed to
handle very large graphs. For example, by sampling from neighborhoods or
loading portions of the graph from storage.

For time-cost scaling, the primary point in the pipeline in which we
suffer scaling challenges is during pre-processing. We build graphs from
the text-file representation of the problems. The time-cost of this step
is linear with the problem size, measured in terms of the number of
clauses. Given this, time-cost is not a major factor in the scaling
issue.

We will add text to clarify the scalability considerations outlined
above and specify the precise nature of an "extremely large" problem
that would challenge our proposed generative model.

\(2\) Public Dataset:

SAT Competition data is an aggregation of thousands of families of
problems and is therefore highly heterogeneous. As discussed briefly in
the introduction, this heterogeneity is highly unfavorable for machine
learning algorithms and tasks. Thus, machine-learning papers are often
required to find creative ways to provide large-scale experimental data
(and this may not conform to real-world data). In \[1\], for example,
data is randomly generated.

In order to demonstrate that our method can generalize to other
datasets, we now provide new results on all UNSAT problems from the SAT
Competition data coming from the "Tseitin" family. The data was compiled
using the GBD database, querying for "family=Tseitin and result=unsat".
There are 135 "Tseitin"-family problems in the database,

We repeat the runtime distribution experiments on this data, generating
5 instances for each of the 135 seed instances. We see very similar
stacked histograms for the solver rankings (Fig. 2 in the attached
response pdf). Fig. 2 (right) shows per-solver boxplots comparing seed
instances and generated instances. Overall these are similar, although
solvers 1 and 6 show greater upper-range in the original data compared
to HardCore. However, the median solve times for solvers 1 and 6 are
still close.

In the table below we show data augmentation experimental results. Since
there are only 135 problems, and our focus is on data-scarce settings,
we use small datasets to train the SATZilla-based predictor. In the
table, we see that for all datasets aside from the smallest one, using
HardCore to generate an augmented dataset leads to a 4%-6% reduction in
MAE compared to training using the original data.

\[1\] D. Selsam et. al., "Learning a sat solver from single-bit
supervision,", in ICLR, 2019.

| Training Data           | 10              | 20            | 30              | 40              | 50              |
|-------------------------|-----------------|---------------|-----------------|-----------------|-----------------|
| HardCore-Augmented      | 3618.9          | **3410.0** | **3311.4** | **3417.7** | **3419.4** |
| Original (Un-Augmented) | **3369.6**| 3581.9        | 3576.1          | 3544.7          | 3608.5          |

   Data Augmentation experiment: MAE of Runtime Prediction averaged
  across 7 solvers and 15 trials. We train a runtime prediction model
  according to the experimental setting in 6.4.2 of the paper. Columns
  in the table indicate the number of original problems used in the
  training set (we generate 4 times per original problem in the training
  set). Results in the "HardCore" row are MAE for a runtime prediction
  model trained on HardCore-augmented data, whereas "Original" indicates
  un-augmented performance.

---

### Decision · Program_Chairs · 2024-09-25

**Decision:**

Accept (poster)

**Comment:**

This paper proposed a new way of synthesizing hard UNSAT formulas (specifically the industrial satisfiability problem instances) for training deep learning methods for SAT solving. The main idea is to have a fast UNSAT core detection procedure using GNNs. With the augmented data, the paper is able to achieve significant improvement, while also to generate instances faster than previous ones.

Overall the reviewers like the balance achieved by the authors between the quality of the generated instances and the speed. And this can be very practically applicable, given the ease of the implementation. While the reviewer vCMn had some initial concerns, the authors have provided convincing rebuttal to address that.

With above, we recommend the acceptance of this paper.